# Risk-Based Ring Vaccination: A Strategy for Pandemic Control and Vaccine Allocation

Dinh Song An Nguyen
The Ohio State University
Columbus, Ohio, USA
nguyen.2687@osu.edu

Marie Charpignon
MIT
Cambridge, Massachusetts, USA
mcharpig@mit.edu

Kathryn L Schaber
Boston's Children Hospital, Harvard
Medical School
Boston, Massachusetts, USA
kathryn.schaber@childrens.harvard.edu

Maimuna Shahnaz Majumder[*]
Boston's Children Hospital, Harvard
Medical School
Boston, Massachusetts, USA
maimuna.majumder@childrens.harvard.edu

Andrew Perrault[*]
The Ohio State University
Columbus, Ohio, USA
perrault.17@osu.edu

## Abstract

Throughout an infectious disease crisis, resources that can be used to slow and prevent spread are often scarce or expensive. Designing control policies to optimally allocate these resources to maximize objectives is challenging. Here, we study the case of ring vaccination, a strategy that is used to control the spread of infection by vaccinating the contacts of identified infected individuals and their contacts of contacts. Using agent-based modeling to simulate an Ebola outbreak, we introduce a risk-based ring vaccination strategy in which individuals in a ring are prioritized based on their relative infection risks. Assuming the risk of transmission by contact type is known and a fixed supply of vaccine doses is available on each day, we compared this strategy to ring vaccination without prioritization and randomized vaccination. We find that risk-based ring vaccination offers a substantial advantage over standard ring vaccination when the number of doses are limited, including reducing the daily infected count and death count, and shifting the pandemic peak by a considerable amount of time. We believe that control policies based on estimated risk can often offer significant benefits without increasing the burden of administering the policy by an unacceptable amount.

*Keywords:* agent-based modeling, ring vaccination, Ebola, public health

[*]These authors co-supervised this research.

**ACM Reference Format:**
Dinh Song An Nguyen, Marie Charpignon, Kathryn L Schaber, Maimuna Shahnaz Majumder, and Andrew Perrault. 2023. Risk-Based Ring Vaccination: A Strategy for Pandemic Control and Vaccine Allocation. In *epiDAMIK 2023: 6th epiDAMIK ACM SIGKDD International Workshop on Epidemiology meets Data Mining and Knowledge Discovery, August 7, 2023, Long Beach, CA, USA.* ACM, New York, NY, USA, 6 pages.

## 1 Introduction

Designing control policies for infectious disease outbreaks can be challenging for several reasons, including scientific uncertainty surrounding newly emerging diseases, many objectives that can be in tension with each other, and limited access to labor and other critical resources. In this paper, we consider the case of *ring vaccination*, a vaccination delivery strategy that is employed when the supply of vaccines and the labor required to administer them is limited. Ring vaccination vaccinates individuals within a *ring*, contacts and contacts of contacts of an infected case. Given a vaccine with appropriate properties, especially the ability to safely inoculate an individual who has been recently exposed, ring vaccination can be highly effective. It has been used as a key tool in several Ebola and smallpox outbreaks [2, 6, 7].

Ring vaccination functions by targeting individuals who would be at a higher level of risk of developing the infection, relative to the general population. For example, in the (early/late) stages of Ebola outbreak of Gulu district, Uganda in 2000, the attack rate across the population was roughly 0.126% [12]. However, the secondary attack rate (SAR), defined as the probability that an infection occurs among susceptible people within a specific set of contacts, can better reflect the relation between social interactions and transmission risk [10]. Yang et al. [15] estimate its value at 2.5%—thus, a vaccine administered immediately after exposure would be about 20 times more effective compared to a randomly delivered vaccination.

However, not all individuals in a ring have the same infection risk. For instance, contacts of contacts are less likely, on average, to become infected because transmission must occur twice. Many observable and unobservable factors may contribute to this risk, including the type and duration of contact between individuals, biological differences that make some people more effective transmitters, multiple exposure paths, and behavioral differences that are caused by the presence or absence of public health monitoring (i.e., immediate self isolation at symptom onset).

Like other control policies that target individuals with elevated risk such as contact tracing, ring vaccination faces a fundamental challenge that the number of such individuals is roughly linear in the number of infected individuals, which varies by orders of magnitude throughout a crisis, but the amount of supplies and labor available per day is roughly fixed. We argue that control policies can leverage estimated risk to prioritize vaccine dose allocation, yielding better performance when supplies are scarce. To that end, we propose a risk-based ring vaccination strategy that leverages the differing risks associated with different contact types, information that can be easily elicited as part of contact tracing.

We evaluate the risk-based ring strategy in an agent-based model (ABM) and consider Ebola as the case study because of its unique transmission intensity bases on type of contact. We show that, when doses are highly restricted, risk-based ring vaccination yields significant benefits over standard ring vaccination and randomized vaccination by not only reducing overall transmissions and deaths but also shifting the pandemic peak. We find that the extra risk associated with ring membership is quickly diluted as there are many more contacts of contacts than contacts, and most contacts have little transmission chance associated with them.

## 2  Agent-based model

We develop an ABM for Ebola Virus Disease (EVD) with $N = 14652$ agents (Table 1). We model two agent characteristics that influence spread and mortality: age and household membership. We replicate the household structure and age distributions from Dodd et al. [5], who collected data in Zambia and South Africa in 2005-2006, and again in Zambia in 2011. Each agent is in one of the six following discrete states on each day: Susceptible ($S$), Incubating ($IC$), Infectious ($I$), Vaccinated but not yet immune ($V$), Deceased ($D$), and Removed (immune or recovered) ($R$). State $S$ comprises agents who have not yet received a vaccine or become immune. State $I$ comprises agents who are capable of transmitting EVD to their contacts who are currently in $S$. At the end of their infectious period, agents in state $I$ transition into state $D$ or state $R$, depending on $Pr(D|\text{age})$. We estimate the age-specific probability of death using previously reported case fatality rates (CFR) of EVD for different age groups [14].

Contacts are sampled daily. We sample household and non-household contacts separately. We assume that contacts between each pair of individuals within a household occurs every day. Non-household contacts are sampled from the population according to the inter-household contact matrix from Ozella et al. [13], collected in a village in rural Malawi, accounting for the age of the person. We assume that the number of contacts follows an independent Poisson distribution for each age-age contact pair.

Each contact has an associated exposure type. For household contacts, we use and sample the exposure types and their distributions observed by Bower et al. [1], which include handling fluids, direct and indirect wet and dry contacts, and minimal to no contact. Direct contact refers to situation in which individuals come into direct contact, such as touching and caring for a patient diagnosed with EVD, whereas an indirect contact refers to situations such as washing clothes or sharing the same bed with an EVD positive patient. In addition, wet contact refers to contact with an EVD patient that is symptomatic (e.g. vomiting, bleeding, etc.) while dry contact refers to contact with patients without any symptoms. Each type of contact associates with a different risk level. For example, a direct contact with fluids is associated with a higher risk of transmission than a dry, physical contact. We let $W_{x,y,t}$ represent the risk ratio of the contact between agents $x$ and $y$. For household contacts, it is the age-adjusted risk ratio from Bower et al. [1]. For non-household contacts, we assign the same type to each, with a risk ratio we set to match with the non-household SAR reported in Dixon et al. [4] (see Inferred parameters). $W_{x,y,t} = 0$ if no contact occurred.

We define the probability of transmission from agent $x$ to agent $y$ on day $t$ as

$$Pr(\text{base}) \cdot W_{x,y,t}$$

where $Pr(\text{base})$ is an inferred baseline probability of infection. The process for inferring this parameter is described in the next section.

***Vaccination.*** The 2017 Guinea ring vaccination trial demonstrates that the vaccine we considered in our simulations (rVSV-ZEBOV) is safe to administer to individuals who are incubating, but do not yet show symptoms [6]. Moreover, rVSV-ZEBOV has 100% effectiveness if administered after exposure. Therefore, we assume that agents in state $IC$ and $S$ are eligible for vaccination. After vaccination, they transition to state $V$, and nine days later, they transition to state $R$, where agents are considered immune.

***Inferred parameters.*** We need to infer the parameters $Pr(\text{base})$ and $RR(\text{non-household})$, the non-household risk ratio, from data. $Pr(\text{base})$ can be interpreted as the probability of transmission for a household contact of the minimal contact type. We set this value in order to match the secondary attack rate (SAR) of the ABM to the SAR that was

**Table 1.** Parameters for the ABM.

| Parameters | Values | References |
|---|---|---|
| *Ebola dynamics* | | |
| Incubation period | Lognormal: $\mu = 2.446$ days, $\sigma = 0.284$ | Legrand et al. [9] |
| Infectious period | Lognormal: $\mu = 2.2915$ days, $\sigma = 0.1332$ | Legrand et al. [9] |
| Case fatality rate | Ages < 15: 77.8% | Qin et al. [14] |
| | Ages 15 - 59: 85.87% | |
| | Ages > 59: 95.7% | |
| Time from vaccination to immunity | 9 days | Kucharski et al. [8] |
| Household secondary attack rate | 12.3% | Dixon et al. [4] |
| Non-household secondary attack rate | 4.8% | Dixon et al. [4] |
| Non-household contact matrix | Adults-Children: Poisson, $\lambda = 1.2$ | Ozella et al. [13] |
| | Adults-Adolescents: Poisson, $\lambda = 1.5$ | |
| | Adults-Adults: Poisson, $\lambda = 5.3$ | |
| | Adolescents-Children: Poisson, $\lambda = 2.0$ | |
| | Adolescents-Adolescents: Poisson, $\lambda = 3.6$ | |
| | Children-Children: Poisson, $\lambda = 0.2$ | |
| *Inferred model parameters* | | |
| Base probability of transmission | 0.01962 | Inferred from Bower et al. [1] |
| Contact type distribution (household) | Handled fluids: 16.3%, $RR : 9.7$ | Bower et al. [1] |
| and risk ratios (RR) | Direct wet contact: 40.3%, $RR : 8.3$ | |
| | Direct dry contact: 17%, $RR : 5.6$ | |
| | Indirect wet contact: 2.6%, $RR : 4.9$ | |
| | Indirect dry contact: 10%, $RR : 1.3$ | |
| | Minimal contact: 13.8%, $RR : 1$ | |
| Risk ratio for non-household | 2.45 | Inferred from Equation 2 |

previously reported for Ebola. Specifically, we solve the following equation for $Pr(\text{base})$

$$SAR_{hh} = Pr(\text{base}) \sum_i Pr(i|\text{household contact})RR(i), \quad (1)$$

where $Pr(i)$ is the probability of a contact having type $i$, $RR(i)$ is the risk ratio associated with contact type $i$. This results in $Pr(\text{base}) = 0.01962$. With $Pr(\text{base})$ identified, we can solve for $RR(\text{non-household})$:

$$SAR_{\text{non-}hh} = Pr(\text{base})RR(\text{non-household}), \quad (2)$$

resulting in $RR(\text{non-household}) = 2.45$, an intensity between indirect wet and indirect dry contact.

## 3 Risk-based ring vaccination

In the risk-based ring vaccination strategy, we prioritize the limited vaccine doses to agents within a ring with the highest estimated risks. The estimation strategy for risks needs to be simple and only use information that is easy to observe. Specifically, we propose estimating risks based on contact type and household membership and doing so only within a ring—thus, there are at most two contact events that contribute to any estimated risk. We assume that risks are estimated separately for each ring and that there is no

coordination between rings. Risks are updated for each individual at most once—we update them for contacts of contacts if the contact becomes infected.

We define a ring as the contacts and contacts of contacts of the infected agent. Let $x$ denote the seed case for the ring, $y$ denote a contact of $x$, and $z$ denote a contact of $y$. We define the risk for $y$ as

$$R(y) = Pr(\text{base}) \cdot W_{x,y}, \quad (3)$$

where $W_{x,y}$ is the risk ratio associated with the highest intensity contact between $x$ and $y$ after $x$ developed symptoms, i.e., $\max_t W_{x,y,t}$ with $t$ in $x$'s infectious period. For $z$, we define the risk as

$$R(z|y \text{ is not infected}) = Pr(\text{base}) \cdot W_{x,y} \cdot Pr(\text{base}) \cdot W_{y,z} \quad (4)$$

$$R(z|y \text{ is infected}) = Pr(\text{base}) \cdot W_{y,z}, \quad (5)$$

using equation 4 if $y$ is not known to be infected and updating to use equation 5 if $y$ becomes infected.

Individuals in the ring are then vaccinated in order of their risk ranking, i.e., each day the $U$ unvaccinated individuals who do not have symptoms with highest risk are vaccinated. If there are still some vaccines left after everyone in the ring has been vaccinated, which can happen when individuals are

unreachable during the vaccination process or in the later stage of the outbreak, then the remaining vaccines will be randomly distributed to the susceptible agents that are not in the identified clusters.

## 4    Preliminary results

We compare the risk-based ring vaccination approach to three baselines: random vaccination, full ring vaccination, and no prioritization ring vaccination. All baselines vaccinate only individuals that have no symptoms and are unvaccinated (i.e., individuals in states $S$ and $IC$). In *random vaccination*, $U$ individuals are vaccinated at random each day. In *no prioritization ring*, $U$ individuals that are in a ring are vaccinated and any leftover vaccines are randomly distributed. In *full ring*, *all* individuals in a ring are vaccinated, relaxing the constraint of $U$ vaccines per day. In all cases, each individual has a 30% to be unreachable (as in [8]). The dose that would go to that individual instead goes to the next eligible agent (i.e., the next highest risk in risk-based or another agent in the ring in no prioritization ring). We simulate the ABM with 10 seed cases selected uniformly at random from the population.

By ranking individuals who are at most at risk, risk-based ring vaccination substantially reduces the infected number of infections and deaths (Fig. 1 and Tab. 2). However, the impact of risk-based prioritization varies significantly across dose limits. In all dose limits, we see a statistically significant difference between risk-based prioritization and standard ring vaccination. This difference is most salient for moderate dose limits—for 100 daily doses, risk-based reduces deaths by roughly 2 times that of randomized vaccination and 1.8 times for no prioritization ring. With 200 doses available, both risk-based and no-prioritization ring differ substantially from randomized vaccination, whereas in 50 and 100 doses, no prioritization ring and random achieve relatively similar performance. In the case of 50 daily doses, risk-based ring has a smaller impact on the number of infections and deaths (< 9% relative to random). However, we see substantial shifting of the infection curve in this setting, delaying the peak by about 20 days.

The full ring strategy (without dose limit) results in few deaths as the vaccine for EVD is highly effective even when administered after exposure, even when 30% of contacts are unreachable at the time of vaccination. However, the cost of this performance is the need for a surge of vaccination in the first month of 321 ± 179 doses per day. This approach achieves control early resulting in an average of 111 ± 152 daily doses across the whole period.

## 5    Discussion and Future Work

Creating control policies during an outbreak is challenging due to resource constraints such as limited healthcare personnel and medical supplies. Using an ABM, we study the impact of ring vaccination strategies under a daily dose limit, and consider EVD as the case study, specifically. We find that, even with vaccination-infection combination that is highly suited to ring vaccination, ring vaccination has limited impact on new infections relative to random vaccination until the number of doses available is sufficiently high. Moreover, the implementation of risk-based ring vaccination we consider only requires slightly more information (contact types), but has an impact even at much lower numbers of delivered doses.

It is expected to observe phase transitions in vaccination programs due to the exponential dynamics involved in infections: when the number of daily vaccine doses passes a threshold, infections will decay exponentially, and the outbreak can be contained. However, this intuition does not apply directly to ring vaccination. Despite the ability of ring vaccination to identify individuals who have a higher risk of infection than the broader population, the impact on new infections is relatively modest. A small modification of standard ring vaccination—involving risk-based prioritization among documented contacts—induces dramatically different behavior. Specifically, for a small number of doses (Fig. 1), a risk-based approach yields a shift in the time at which the peak in new infections is reached, thus postponing a surge more efficiently than standard ring vaccination and randomized vaccination. Moreover, above a certain threshold, lying between 50 and 100 daily doses in our model, benefits of the risk-based approach compound and the shift in the timing of the peak is coupled with a significant reduction in the maximum number of new infections. These two distinct effects and their potential coupling are not well understood and merit further study.

A key question is whether more sophisticated vaccination strategies such as ring vaccination are worth the additional overhead cost of reliably identifying and contact tracing cases. The answer to this question is multi-faceted and will depend on the interplay among outbreak stage, vaccine availability, and the combination of vaccination and infection properties. More effort is needed to understand these interactions: during an infectious disease emergency, resources are scarce and need to be allocated towards the geographical areas or subpopulations that result in the highest impacts, i.e., the largest reduction in the maximum number of new infections and the greatest delay in the timing of the peak.

Our study has several limitations. Our current ABM does not incorporate realistic superspreading dynamics. Yet many infectious diseases demonstrate a high degree of transmission heterogeneity, i.e., relatively few seed cases cause many secondary infections [11]. While not well captured in our model, this aspect has substantial consequences for ring vaccination because the variance of the strategy's outcome is increased, i.e., a single missed secondary case can have a

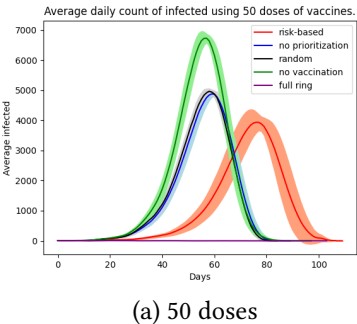

(a) 50 doses

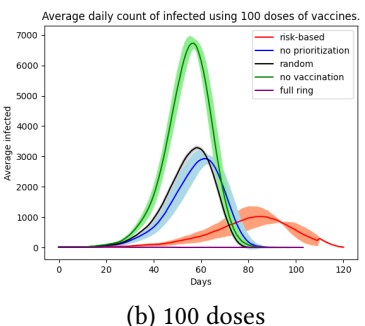

(b) 100 doses

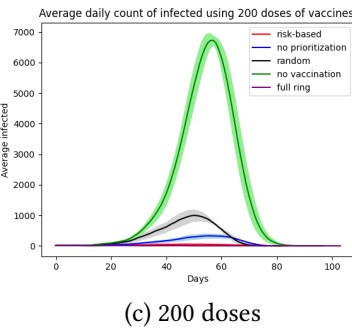

(c) 200 doses

**Figure 1.** The daily mean count (± standard deviation) of infected under different vaccination strategies. We simulate outbreaks with 10 seed cases for each policy given different numbers of vaccine availability. The shaded region indicates the standard deviation for each vaccination strategy.

**Table 2.** Mean (95% CI) count of deceased for each strategy and dose limit.

| Strategy | | 50 doses | 100 doses | 200 doses |
|---|---|---|---|---|
| Risk-based ring | | 8465.77 | 3268.67 | 175.77 |
| | | (8370.63−8560.91) | (1399.83−5137.50) | (144.14−207.4) |
| No prioritization ring | | 9184 | 6091.50 | 784.7 |
| | | (9101.12−9266.88) | (5915.62−6267.38) | (663.08−906.32) |
| Random | | 9272.33 | 6488.57 | 2044.4 |
| | | (9164.44.35−9380.22) | (6425.06−6552.09) | (1627.39−2461.41) |
| Full ring | 27.33 | | | |
| (no dose limit) | (10.79−43.87) | | | |
| No vaccination | 12189.80 | | | |
| | (12156.43−12223.17) | | | |

much larger impact on the timing of the peak in new infections and its magnitude than in the absence of transmission heterogeneity. We suspect that accounting for superspreading events would further reduce the benefits of ring vaccination. However, in some circumstances, pronounced superspreading can make risk-based targeting more effective as observations from a given ring can be used to infer the transmission potential of the seed case.

Furthermore, it is already a hard task to gather contacts and contacts of contacts to form a ring for vaccination. Obtaining information regarding exposure types between infected individuals and their contacts is even more time and resource intensive. Although risk-based ring vaccination is more effective in our results, it is important to consider additional factors like timing and human resources in order to better evaluate the efficacy of our method.

By design, ring vaccination targets individuals with a higher number of contacts or more centrally located in a network. These individuals tend to get infected earlier than their counterparts with an average number of contacts and centrality [3]. *Risk-based* ring vaccination, by prioritizing individuals with contacts at higher risk, will additionally target individuals in larger households. This additional feature operates independently from the "encirclement" aspect of

standard ring vaccination; more work is needed to quantify their respective contributions (e.g., by comparing risk-based vaccination to strategies that prioritize individuals based on household size).

## Acknowledgments

KS was supported in part by grant SES2200228 from the National Science Foundation. MSM was supported in part by grant R35GM146974 from the National Institute of General Medical Sciences, National Institutes of Health. The funders had no role in study design, data collection and analysis, decision to publish, or preparation of the manuscript.

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
