# OpenReview forum: "Risk-Based Ring Vaccination: A Strategy for Pandemic Control and Vaccine Allocation"
_KDD.org/2023/Workshop/epiDAMIK — KDD 2023 Workshop epiDAMIK_

### Official Review · Reviewer_6DTd · 2023-06-27
**Risk-based ring vaccination appears promising. Need more analysis to test robustness (to demographic patterns) and generality (to other infectious diseases)**

**Rating:** 2
**Confidence:** 4

**Review:**

The paper explores preliminary work for risk-based ring vaccination as an intervention to control spread of infectious diseases given limited resources. Authors consider the specific case study of Ebola vaccination and compare with multiple protocols: random-vaccination, no-prioritization ring vaccination, full ring vaccination. Simulations show that risk-based ring vaccination is shown to be promising to achieve full-ring benefits with significantly fewer resources. However, significantly more experiments are need to make strong and reliable inferences.

Some comments/questions for the authors to think about:
1. Risk-based ring vaccination seems very sensitive to mobility patterns and presence of NPIs (like lockdowns). Does this only work when communities are sparse and isolated or with active mobility patterns. How do you form a ring then? Authors should simulate with real-scale populations, with dynamic movement patterns and calibrate with real-world data sources before making interventional claims. How was this model calibrated?
2. Compare with other non-ring based resource-limited vaccination strategies. For instance, during COVID-19: some govts delayed 2nd dose of the COVID-19 vaccine to prioritize first doses; and prioritized high-risk age groups when the supply was limited, such as [1]. Is risk-based ring vaccination better than these methods? Maybe interesting to study in the next paper.
3. Does risk-based ring vaccination also generalize to other infections like COVID-19/Flu which spread at mass scale or is only good when infectious are more localized to smaller communities, like Ebola. Would be important to analyze and clarify this distinction.
4. Finally, a "somewhat" similar concept explored in 106 canada neighborhoods during COVID-19 alpha variant, as in [2]. With the alpha variant, most infections were with <18 yr olds but vaccines were not authorized yet. So, authorities vaccinated parents of children at greater risk from COVID-19 since vaccine was not authorized for children yet. Is this a form of risk-based ring vaccination? The idea is very intuitive, so i am curious to know if such risk-based rings have been explored previously.

I would encourage the authors to think about some of these concerns, if they are selected to present at the workshop. I am also okay if the paper is accepted since it is a non-archival workshop and would make for good discussion.

[1]: https://www.bmj.com/content/373/bmj.n1087
[2]: https://jamanetwork.com/journals/jamanetworkopen/fullarticle/2788978

---

### Official Review · Reviewer_dEvV · 2023-06-28
**This paper proposed a risk-based ring vaccination method that achieves better performance than the existing no-prioritization ring method and random method.**

**Rating:** 4
**Confidence:** 5

**Review:**

This paper proposed a risk-based ring vaccination method that achieves better performance than the existing no-prioritization ring method and random method.

Strength:
1. Good motivation: The idea of risk-based vaccination allows more effective vaccine distribution.
2. The experience showcases the effectiveness of the proposed risk-based ring vaccination method.

Weakness:
1. Only one experiment setup is used in experiments for evaluation. Another experiment for other diseases, or at least one other model, is more useful to better showcase the proposed method.
2. The vaccine budget (50/100/200) seems a little random. A better budget based on real-world Ebola vaccine production rate is more useful to showcase the effectiveness of the proposed method in the application

---

### Official Review · Reviewer_cUSa · 2023-06-29
**Review of the paper on risk-based rink vaccination**

**Rating:** 5
**Confidence:** 5

**Review:**

The paper is written and explained very well. The authors have employed agent-based simulation, incorporating six distinct states and separate sampling for household and non-household contacts. The authors have introduced risk based ring vaccination and showed that it is more effective compared to the random allocation and ring allocation. Furthermore, the authors have provided insightful suggestions for potential future research directions, all of which are highly intriguing and would greatly enhance the existing work.

The assumptions regarding within-household contact appear logical, while estimates for non-household contact draw from a social contact pattern study conducted in Malawi. It's important to talk about why these assumptions and estimates are important and how they affect the proposed vaccine strategy.

It would be interesting to discuss the C.I. patterns shown in Figure 1. Specifically, we could look at whether the variability decreases for certain vaccine allocation strategies after a certain number of days. Notably, in Figure 1(b), why does the curve based on ring vaccination exhibit such a narrow range between 80 and 100 (around 90-95)?

---

### Official Review · Reviewer_ywYu · 2023-06-30
**Review of risk-based ring vaccination**

**Rating:** 4
**Confidence:** 5

**Review:**

In this paper, the authors investigate a risk-based ring vaccination strategy. Ring vaccination is a vaccine allocation strategy that vaccinates the contacts and contacts-of-contacts of an infected case. Here, the authors use an agent-based model to simulate an Ebola outbreak and test a variant of ring vaccination that prioritizes individuals within the contact-of-contact network with the highest risk (with risks estimated from the model). They show through their simulations that risk-based ring vaccination is significantly more effective than ring vaccination without prioritization, especially when more doses (100 or 200) of the vaccine are available.

Strengths
+ Risk-based ring vaccination is a nice idea and well-motivated
+ The authors clearly demonstrate the effectiveness of this strategy through simulations
+ The model is largely motivated by prior literature and uses parameters from prior work

Weaknesses
- The results feel almost like a foregone conclusion given the model, since they use risks from the model to decide which individuals to prioritize. It would be useful to establish, especially through mathematical analysis if possible, if we should be "surprised" by the results, or the settings that must hold true for risk-based to be significantly more effective.
- A lot of design decisions are made within the model, eg, levels of contact and types of contact within households/across households, disease parameters, etc. While it helps that parameters were mostly set based on prior literature, it would be useful to conduct sensitivity analyses to see how model results vary based on the decisions made.
- Unclear if authors were the first to do risk-based ring vaccination. Also, unclear how realistic this model is in real life, since their simulation uses the individual's "real" risk from the model to determine prioritization. In reality, it seems hard already to get an infected person's contacts and contacts-of-contacts; would be even harder to know levels of contact/risk between all these people.